# Fetal yawning and mouth openings: Frequency, developmental trends, and association with birth weight

Damiano Menin[1], Paola Veronese[2], Maria Teresa Gervasi[2], Harriet Oster[3], Marco Dondi[1]*

1 Dipartimento di Studi Umanistici, Università degli Studi di Ferrara, Ferrara, Italy, 2 Department of Women's and Children's Health, Maternal Fetal Medicine Unit, University of Padua, Padua, Italy, 3 School of Professional Psychology, New York University and Department of Psychology Hunter College, City University of New York, United States of America

* marco.dondi@unife.it

## Abstract

During the last 15 years, the brain cooling hypothesis has shown unparalleled explanatory and predictive power among the several attempts aimed at elucidating the phylogenetic origins of yawning. However, some blind spots remain which are not directly accounted for by this theoretical explanation, including the presence of yawning in human fetuses, as their thermoregulation is largely dependent on the mother. However, the few studies which addressed fetal yawning are often plagued by serious methodological issues, in particular concerning the validity and reliability of methods adopted to identify yawns, resulting in contradictory results. In the present study, we scored yawns and other mouth openings in 32 healthy fetuses observed during ultrasonographic scans between the 23rd and the 31st gestational week, using the Baby FACS-based System for Coding Perinatal Behavior (SCPB). We found average yawning frequencies to be below 5 per hour, and not related with gestational age (GA). Non-yawning mouth openings, instead, showed a GA-related decrease that, together with validity issues of measurement methods, might explain the similar developmental trend found for yawning frequencies in two previous studies. Finally, yawning frequencies were negatively related with birth weight, considered as an indicator of mild distress, potentially showing a stress-related modulation of yawning behavior in healthy fetuses.

## Introduction

Yawning is a phylogenetically and ontogenetically primitive behavioral pattern, virtually ubiquitous to vertebrates [1,2], which is observed in human fetuses since the 11[th] gestational week [3] and unchanged throughout life [4].

**Data availability statement:** All relevant data are within the paper and its Supporting Information files.

**Funding:** The author(s) received no specific funding for this work.

**Competing interests:** The authors have declared that no competing interests exist.

The fact that yawning frequencies have been found to be modulated by a vast range of conditions and stimuli, including but not limited to stress [5,6], hunger [7,8], pain [9], arousal [10] and thermoregulation [11], has probably contributed to the proliferation of theories about this behavior's potential functions and phylogenetic origins. In particular, despite the interest that this peculiar behavior has generated in scholars throughout history [12], up until a few decades ago this fascination manifested itself mostly in terms of theoretical speculations and yawning was generally considered as a mechanism to revert hypoxia, despite the lack of empirical evidence supporting this hypothesis [13].

Starting with the 1980s, however, a new interdisciplinary interest towards yawning has risen, leading to a considerable accumulation of evidence regarding the conditions and factors that modulate yawning behavior, as well as the neuropharmacological processes involved, and to the formulation of several hypotheses aimed at explaining these evidences in terms of evolutionary functions [14,15].

The popular opinion according to which yawning is a respiratory maneuver aimed at increasing oxygenation and/or decrease $CO_2$ levels of blood was thoroughly tested and rejected by Provine et al. [13], who found that neither exercise nor experimentally manipulated $O_2$ or $CO_2$ levels in the blood had any effect on yawning frequencies.

During the following decades, the interdisciplinary interest toward yawning has risen, leading to the formulation of several hypotheses about the ultimate causes of this behavioral pattern. In particular, different scholars have proposed several modulating factors as an expression of the original function of yawning, including state change [16,17], arousal regulation [18], cortisol levels and stress regulation [19], empathy and social interactions [20,21] and brain thermoregulation [22].

Although these hypotheses are still proposed and argued for as alternative explanations by some scholars [21,23], in the last 15 years, the brain cooling hypothesis, according to which the phylogenetically original function of yawning lies in its ability to regulate brain temperatures, has been gaining notable empirical support. This perspective offers a potentially unifying explanation for much of the evidence related to different yawning modulation mechanisms [11,24]. In fact, the brain cooling hypothesis is not only corroborated by evidence pertaining to the conditions and mechanisms involved in yawning modulation, but has been substantiated with data showing that the behavioral pattern involved in yawning actually produces the effect hypothesized, i.e., reducing brain temperatures when the environmental temperature allows it [25]. It is important to note, however, that evidence of this modulation is currently available only for rats [25], budgerigars [26] and mice [27], and additional studies are needed to confirm these results in other species. This thermoregulatory function of yawning has also been plausibly explained based on the circulatory changes accompanying yawning episodes in all homeotherms [11,28]. In general, a striking number of predictions based on the brain cooling hypothesis were later confirmed, based on observational and experimental data, e.g., regarding the thermal window where yawning is more often observed [29], the positive correlation between yawn duration, brain size and neuron numbers across mammals and birds [2] and the modulation of yawning Icontagion due to neck temperature manipulation [30].

Moreover, advocates of the brain cooling hypothesis have been able to more parsimoniously account for an impressive amount of evidence originally presented in the context of other theoretical frameworks. In fact, many proximate causes for yawning can be explained in terms of an underlying thermoregulatory function, including state changes, stress and drug-induced modulation [11,24].

Overall, to the best of our knowledge, despite the ongoing discussion, the brain cooling hypothesis has shown unparalleled explanatory and predictive power among the several attempts aimed at elucidating the phylogenetic origins of yawning and at tracing back different forms of modulation to a singular core function. However, some blind spots remain which, albeit not necessarily falsifying the brain cooling hypothesis, are not directly accounted for by this theoretical explanation, at least in its original or current forms.

The most widely investigated and discussed of these blind spots concerns contagious yawning, a phenomenon which has been observed in some highly social species, including humans and several other primates, dogs, captive wolves and pigs [21,24]. In particular, although the contagiousness of yawning has been explained in terms of it being a cue providing information about the reduced alertness of the yawner, rather than a communicative signal [24,31], some (albeit limited) evidence seems to indicate that familiarity, empathy and/or emotional contagion might play a role in facilitating yawning contagion [21]. The fact that familiarity and emotional closeness seem to increase the likelihood of yawning contagion has therefore led to to the formulation of two alternative hypotheses: the *Attentional Bias Hypothesis* and the *Emotional Bias Hypothesis*. The former considers the effect of familiarity on yawning contagion as being mediated by the time spent looking at the yawner's face and attentional focus [32], while the latter regards it as a direct effect of positive social bonding [21,33] and, if confirmed, might suggest that, at least for the species for which yawning can be contagious, there are dynamics of yawning modulation that can not be traced back to brain thermoregulation. Interestingly, proponents of the *Emotional Bias Hypothesis* argue that yawning contagion might be an exaptation of this plesiomorphic behavior to facilitate emotional contagion [33]. This would support the idea that this largely conserved behavioral pattern could be serving at least partially different evolutionary functions across species.

Another blind spot of the brain cooling hypothesis is that it only applies to homeotherms [11]. However, as Gallup himself recognized [24], considering the ubiquity of yawning among vertebrates, it is likely that this behavior originally evolved in jawed fish. This seems to indicate that the thermoregulatory function of yawning may not have been present at its phylogenetic origin, and might be hypothesized to derive from one or more more primitive functions. Yawning in fish and other poikilotherms is still a vastly understudied phenomenon, but a recent study has found that white-spotted chars show higher frequencies of yawning immediately before a behavioral transition from stationary to active [34], suggesting that the mechanism of yawning modulation related to state changes can be observed in species where it cannot (at least, to the best of our knowledge) be explained in terms of thermoregulation.

Another phenomenon which eludes explanation in terms of the brain cooling hypothesis is fetal yawning. Because fetal thermoregulation is largely, although not solely [35] dependent on the mother, Walusinski [36] has even argued that the brain cooling hypothesis overlooks the very existence of fetal yawning. However, as Gallup & Eldakar [11] pointed out, this argument relies on the assumption that any behavior that can be observed in utero serves the same function(s) after birth.

Nevertheless, similarly to the phenomena related with contagious yawning and yawning in poikilotherms, although the existence of fetal yawning might well be compatible with the brain cooling hypothesis, it is still a challenge that should be addressed in order to pursue an organic and systematic theory of the phylogenetic and ontogenetic origins of yawning. In particular, if similarly to what observed by Yamada & Wada [34] in white spotted chars, fetuses were to yawn in somewhat similar conditions to homeotherms, this might indicate that there is one or several more primitive function(s) of yawning, or, alternatively, that yawning somehow serves a thermoregulatory function also in human fetuses and in poikilotherms, as hinted at by Gallup & Eldakar [11].

Despite the theoretical relevance of this topic in the context of yawning research, however, fetal yawning is still somewhat overlooked. This lack of attention is probably due to the practical challenges inherent in studying behavior in utero

based on US scans, often characterized by low spatial and temporal resolution. Moreover, the few existing studies have yielded somewhat contradictory results, e.g., regarding the estimated yawning frequencies across fetal development, ranging from zero [37,38] to over 10 [37,39,40] yawns per hour across the third trimester of pregnancy.

As Menin et al. [41] pointed out, this variability is likely due to issues regarding the validity and reliability of the methods adopted in order to identify yawns. In fact, most studies only employed a single coder and therefore presented no estimate of inter-rater reliability, while at the same time adopting very concisely worded descriptions or even video samples exemplifying the behavioral pattern as observational measurement tools [41]. One study [37], on the other hand, adopted a method based on the temporal dynamics of mouth movements, classifying as yawns all of the mouth openings with an opening phase longer than their closing phase. This pioneering approach, which identified yawns based on an objective timing-based criterion, resulted in good reliability, but was later showed to have very limited specificity (resulting in a high number of false positives) in a study [41] adopting a detailed description based on Baby FACS [42] as benchmark in a preterm neonate model. In particular, Menin et al. [41] showed that, while their method resulted in classifying 11.5% of mouth openings as yawns, the one adopted by Reissland et al. [37], in line with their original results, led to identifying 67.5% of mouth openings as yawns.

These validity issues highlighted in the method based on the opening-to-closing duration ratio [37] raise some questions as to the soundness of the findings of their study, first of all the sharp decrease associated with gestational age (GA) that authors of that study found both in yawning and non-yawn mouth opening frequencies. This doubt is even more relevant, as the study by Reissland et al. [37] is one of the few to investigate the fetal development of yawning frequencies, and is relatively widely cited as evidence that yawning may be used as an index of fetal development [43–45].

To the best of our knowledge, the result regarding a supposed GA-related decrease in yawning frequencies, was only partly replicated by another study [46], where authors found only a small correlation between GA and yawning frequencies. This study, however, may suffer from the same psychometric issues that were also described by Kurjak et al. [47], as it employed a single coder and used a relatively generic description which, similarly to the method based on the opening-to-closing duration ratio [37], risks classifying far too many mouth openings as yawns. Other studies [47–49] did not find such an effect of GA.

### Research questions and hypotheses

The present study aimed at shedding some light on one of the blind spots in the brain cooling hypothesis, namely the one concerning fetal yawning, by addressing the following research questions:

(1) What are the average yawning frequencies over the second and third trimester of gestation, as estimated using valid and reliable measurement methods?

(2) Can the sharp GA-related decrease in yawning frequencies found by Reissland et al. [37] and AboEllail et al. [46] be confirmed using such measurement methods?

(3) Can a similar developmental trend be found for non-yawning mouth openings? Could this trend explain the differences previously attributed to yawning?

(4) Is the average duration of yawning and non-yawning mouth openings associated with GA or birth weight?

(5) Are fetal yawning frequencies related with variables potentially associated with yawning modulation in extra-uterine life (e.g., stress-related variables)?

In order to tackle this last issue, as our aim was to investigate potential non-pathological modulators of yawning rates, instead of comparing high-risk and low-risk fetuses as done, e.g., in Petrikovsky et al. [50], we used birth weight as a predictor of yawning and non yawning mouth opening rates in fetuses that were later born full-term and with a weight appropriate for

gestational age (AGA). Slightly low birth weight, in fact, was found to be associated with increased short-term and long-term morbidity [51] and increased stress [52,53] and even higher risk of cognitive impairments [54,55]. We therefore considered that a slightly reduced birth weight might be associated with moderate distress during the third trimester of pregnancy.

We hypothesize that the median and mean yawning frequencies in fetuses over the second and third trimester are lower than 5 yawns per hour (H1), similar to those observed in preterm neonates [8,56], and do not decrease nor increase with GA (H2). We also hypothesize that non-yawning mouth openings show a GA-related decrease, which, together with the use of non-specific measurement methods, would explain the developmental trend found by Reissland et al. [37] for yawns (H3).

Moreover, because the duration of a mouth opening episode is one of the cues often adopted to identify yawns, we hypothesize that the average duration of non-yawning might show a GA-related decrease (H4) which would partially explain the effect found by Reissland et al. [37]. Finally, we hypothesize that yawning frequencies might be negatively related with birth weight, considered as an indicator of mild distress (H5).

## Materials and methods

### Participants

Thirty-two healthy fetuses (18 females, 56%, 14 males, 44%) were scanned between the 23rd and the 31st gestational week (M = 26.17, SD = 2.03). All clinical and developmental markers were within normal reference range, and all fetuses, examined after birth, were found to be healthy and were born full-term. All of them had a birth weight appropriate for gestational age, between 2.5 kg and 4 kg (M = 3.23, SD = 0.34) and had an Apgar score of nine or ten at one and five minutes.

Maternal exclusion criteria were: diabetes type I or II, smoking, alcohol or substance abuse. Pregnancy-related exclusion criteria were gestational diabetes, pregnancy-induced hypertension, Rh immunization, placental bleeding, fetal anomalies and polidramnios. Fetal exclusion criteria were congenital malformations, intra-uterine growth restriction (IUGR) and multiple pregnancy. Pregnant women who had children with adverse neurological outcomes, preterm deliveries, recurrent spontaneous abortions or other complications in their obstetric history were also excluded. Finally, we excluded fetuses who later showed health problems at birth, a non-optimal Apgar score at one and five minutes (less than 9 and 10 respectively), those who had to take medications and those born preterm. This choice was based on previous exclusion criteria present in the literature [40,57–60]. All mothers were of Italian nationality, with a self-reported Italian ethnic background.

Ultrasonographic scans were performed at the Complex Operating Unit (UOC) of Obstetrics and Gynecology of the Padua Hospital (Italy) between January 3, 2013 and March 13, 2015. The Ethics Commission of the Padua Hospital approved the experimental design (IRB number: 1381P) and a written informed consent was obtained from all the participating pregnant women.

### Procedure

Ultrasound examinations were performed by an expert gynecologist, using a Voluson 730 ultrasound machine (Expert GE Healthcare) with a 5 MHz trans-abdominal transducer. In particular, the Voluson 730 can acquire volumetric images in real time up to 40 volumes for second. It allows real-time reconstruction of the anatomical volume, showing precise and quantifiable information that can be used for diagnosis. In order to investigate fetal behavior, 4D images were obtained thanks to the automatic scanning of the body volume. The image sequences were recorded on DVD.

In the two hours preceding the examination, the pregnant women had not eaten and scans were conducted in the early afternoon while they were in the supine position. The mean duration of the recordings was 22.50 minutes (SD = 10.00).

### Behavioral coding

Frame by frame behavioral analysis of video-recordings was performed by two independent expert FACS and Baby FACS coders (with the secondary coder examining 30% of the videorecordings, randomly selected, n = 10), using ELAN, a professional software [61] for the creation and management of complex annotations on video and audio (Max Planck Institute

for Psycholinguistics, The Language Archive, Nijmegen, The Netherlands; http://tla.mpi.nl/tools/tla-tools/elan/). All codings were carried out at the Early Infancy Lab of the University of Ferrara (Italy).

## Mouth openings coding

In line with Reissland et al. [37] and Menin et al. [41], mouth openings were coded when Action Units (AUs) 25 (*lips parted*) and 27 (*mouth stretch*), as described in Oster's Baby FACS [42], were simultaneously observed, resulting in the mouth being stretched widely open and the mandible being pulled down vertically.

## Yawn coding

After having identified mouth openings, coders classified each event either as a yawn or a non-yawn mouth opening, according to the following description from *The System for Coding Perinatal Behavior* (SCPB) [62], based on the AUs described in the Baby FACS [42] and previous studies in the literature [3,18]. Recent studies used SCPB to identify yawning and other behavioral patterns in fetuses [41], preterm neonates [8,63] and infants [6].

Yawning is defined in the SCPB as a stereotyped behavior characterized by a slow mouth opening with deep inspiration, followed by a brief apnea and a short expiration and mouth closing, typically accompanied by limb stretching [3]. The expansion of the pharynx can quadruple its diameter, while the larynx opens up with maximal abduction of the vocal cords [18]. One of the characteristic features of yawning [42] is its timing, consisting in a progressive acceleration, followed by an abrupt deceleration in the intensity of the facial muscle Action Units (AUs) involved, designated by numeric codes and verbal labels. Yawning usually emerges from a relaxed face, initially involving mouth opening (AUs 25, 26, 27) and eyes closing (AU 43E), followed by upper eyelid drooping (AU 43A-D), flattened tongue on the bottom of the mouth (AU 75) and usually swallowing (AU 80). During the plateau brow knitting (AU 3), brow knotting (AU 4), nose wrinkling (AU 9), lateral lip stretching (AU 20), nostril dilation (AU 38) and head tilting back (AU 53) also typically occur.

## Data analysis

We used Cohen's Kappa to assess inter-rater reliability, adopting a one-second threshold for the individuation of both onsets and offsets of mouth openings. The analysis showed good reliability both for the identification of mouth openings (Kappa = .88) and for their classification as yawns or non-yawn mouth openings (Kappa = 1.00).

The observation time was identified for each scan by excluding from the total duration of the recording the time intervals in which the visibility of the fetus' face was not sufficient to identify the behavioral patterns of interest.

We performed a series of regressions in order to investigate the potential relationships between gestational age and birth weight on the one hand and the occurrences of yawns and non-yawn mouth openings on the other hand. In order to accommodate for the discrete count nature of the data, the analyses pertaining to the occurrences of yawns and non-yawn mouth openings were performed using Poisson regressions. The observation time was used as a logarithmic offset to account for the variability in the length of each scan. The mean number of yawns per hour was calculated for yawns and mouth openings solely for the purpose of performing descriptive analyses to provide a clear picture of the frequencies of observed behaviors. Although fetal sex was included in preliminary analyses as a predictor, it was dropped from the final regressions because it did not significantly improve the model fit (tested via ANOVA) and showed no significant correlation with other fixed effects. This decision is further supported by the current lack of evidence for sex-based modulation of behavior in fetuses, a population showing minimal sexual differentiation. The complete results of the full models and the ANOVA, however, are provided in the Supporting Information.

Gestational age and birth weight were used as independent variables for all regressions. To determine the statistical significance of the tests used, a conventional alpha value of .05 was assumed. All analyzes were performed using the stats package in R, version 4.2.1.

## Results

### Descriptive statistics

The mean observation time across ecographic scans was 17.4 minutes (SD = 7.3). The number of yawns observed for each video ranged from zero to six, with a mean frequency per hour of 3.63 (SD = 4.64, Mdn = 2.11). The mean duration of yawning observed in individual fetuses ranged from 2.03 seconds to 5.68 seconds (M = 3.58, Mdn = 3.44, SD = 1.01). The number of non-yawning mouth openings observed for each scan ranged from zero to 23, with a mean frequency per hour of 15.69 (SD = 15.28, Mdn = 12.07). The mean duration of non-yawning mouth openings observed in individual fetuses ranged from 0.77 seconds to 7.25 seconds (M = 3.13, Mdn = 2.92, SD = 1.53).

### Regressions

**Yawning.** The Poisson regression with the number of observed yawns as the dependent variable and GA and birth weight as the independent variables, accounting for the observation time as logarithmic offset, showed a negative relationship between the number of yawns and birth weight (see Fig 1), $F_{(2, 29)} = 4.938$, $\beta = -0.420$, $p = .028$, while no association was found between the number of yawns and gestational age, $F_{(2,29)} = 0.170$, $\beta = 0.047$, $p = .822$ (see Fig 2).

The mean duration of observed yawns, analyzed using a linear regression, showed no significant relationship with gestational age, $F_{(2, 15)} = 0.682$, $\beta = -0.252$, $p = .442$ or birth weight, $F_{(2, 15)} = 0.003$, $\beta = 0.001$, $p = .956$.

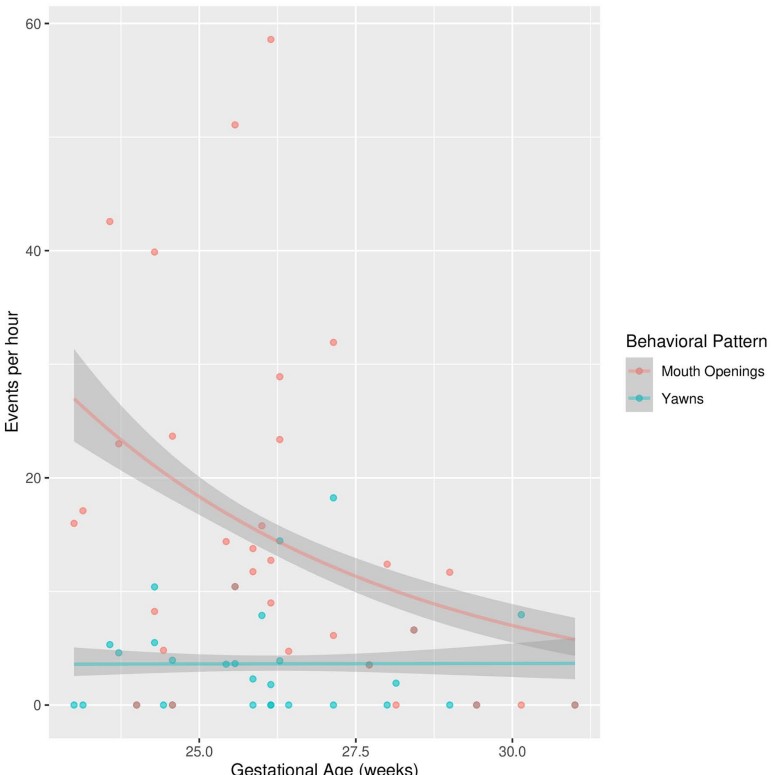

**Fig 1. Yawns and non Yawns Mouth Openings per Hour by Birthweight.** Observed and fitted yawns and non-yawning mouth openings per hour by birthweight with 95% confidence interval.

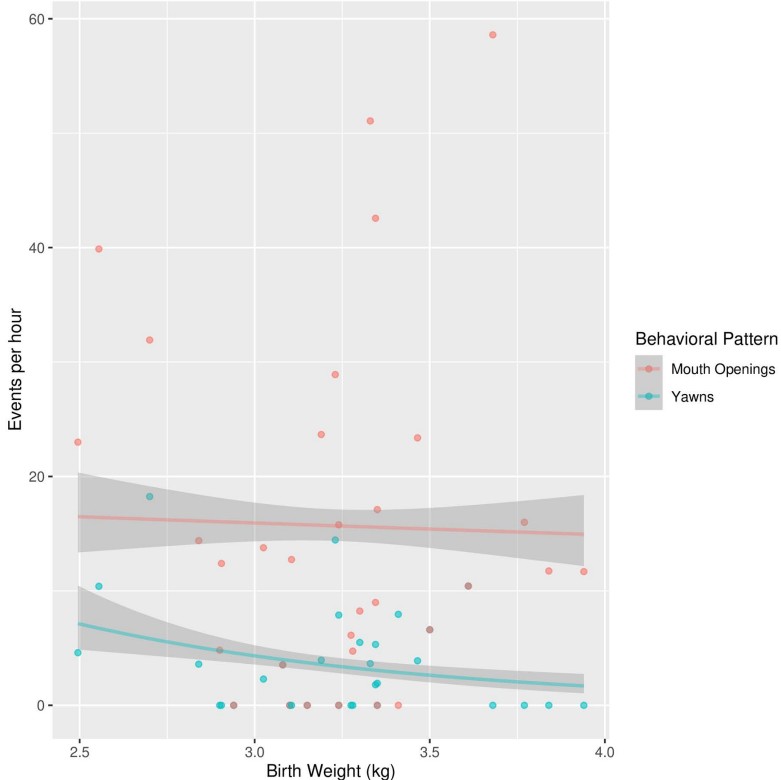

**Fig 2. Yawns and non Yawns Mouth Openings per Hour by Gestational Age.** Observed and fitted yawns and non-yawning mouth openings per hour by gestational age with 95% confidence intervals.

## Mouth openings

The number of mouth openings per unit of time, analyzed using a Poisson regression similar to that used for yawns, showed a negative relationship with gestational age, $F_{(2,29)} = 25.175$, $\beta = -0.480$, $p < .001$ (see Fig 2), while no association emerged with birth weight, $F_{(2,29)} = 1.480$, $\beta = -0.001$, $p = .223$ (see Fig 1).

The mean duration of observed mouth openings, analyzed using linear regression, revealed a negative relationship with gestational age, $F_{(2,23)} = 4.349$, $\beta = -0.833$, $p = .035$, while it did not show associations with birth weight, $F_{(2,23)} = 4.938$, $\beta = 0.001$, $p = .288$.

## Percentage of yawn mouth openings

Finally, for each participant, we calculated the percentage of yawns with respect to the total number of mouth openings (yawns included). A linear regression showed a positive relationship of this variable with gestational age at the time of observation, $F_{(2,25)} = 8.118$, $\beta = 0.163$, $p = .007$, while no associations with birth weight were found, $F_{(2,25)} = 0.493$, $\beta = -0.001$, $p = .489$.

## Discussion

This study aimed at addressing some research questions relevant to the overall study of yawning, as well as some specific to fetal behavior.

First, despite a high variability, which might affect estimates especially when considering small samples and relatively brief observation times, our results confirmed that the average frequency of yawning over the second and third trimester of gestation is significantly below five yawns per hour (see H1) [8,48,56]. In particular, the median frequency we found was around two yawns per hour.

This finding corroborates the hypothesis that studies which reported higher frequencies have likely used non-specific measurement methods, resulting in the classification of many non-yawning mouth openings as yawns. The frequency we estimated is similar to that found in preterm neonates [8], and was not associated with GA (see H2). This, together with the GA-related decrease we found for the frequency of non-yawning mouth openings (see H3), provides further evidence for the hypothesis that the developmental trend highlighted by Reissland et al. [37], as well as the high frequencies they estimated at the beginning of the third trimester of gestation, are due to the adoption of a non-specific method for identifying yawns among mouth openings episodes [41]. Furthermore, the GA-related decrease we found for the average duration of non-yawning mouth openings (see H4), might have contributed to the previous partial replication of this developmental trend by AboEllail et al. [46], as longer mouth openings, in absence of a reliable coding method, might be more likely to be erroneously classified as yawns.

Finally, we found a negative association between yawning frequencies and birth weight (see H5), which represents, to the best of our knowledge, the first evidence of yawning modulation in healthy fetuses. The fact that high yawning frequencies were a predictor of low birth weight may have some implication in terms of fetal neurobehavioral assessment, potentially allowing practitioners to anticipate slightly problematic outcomes even for full-term pregnancies within the physiological range for birth weight, and potentially to predict instances of slightly low birth weight. Moreover, in terms of yawning research, this result is particularly relevant, as it seems to highlight a form of stress-related modulation similar to the one observed in extra-uterine life for humans and other homeotherm species.

Overall, these findings offer a picture of fetal yawning as closer in frequency and modulatory dynamics to extra-uterine yawning than previously thought. In particular, fetuses showed similar yawning frequencies to those observed in neonates and infants, and highlighted a seemingly stress-related [5,64] modulatory mechanism (namely, the negative association with birth weight). The discrepancy between our findings and those of previous studies [37,46], aligning with the methodological concerns raised by Menin et al. [41], underscores the importance of a renewed attention towards the validity and reliability of methods used to identify yawns, especially but not exclusively in fetuses, where limited ultrasound video quality can complicate accurate identification.

In terms of the overarching theoretical discussion on the functions of yawning throughout life and across different species, the evidence regarding the relationship between yawning frequencies and birth weight is compelling but far from conclusive: the seemingly stress-related modulation observable in healthy fetuses, in fact, provided that it is confirmed by later studies, might be due to some original function ontogenetically and maybe also phylogenetically preceding the thermoregulatory one. Alternatively, this evidence might indicate that yawning somehow serves a thermoregulatory function even in fetuses, but it might even not serve any evolutionary function and just represent a byproduct of the thermoregulatory function that can be observed after birth.

It should be noted as a limitation that this study did not employ experimental manipulation or selective observation of fetuses under specific conditions, and did not include measures (e.g., maternal body temperatures or fetal heart rate) that could be useful in disentangling the effects of different modulatory mechanisms, with particular regard for the thermoregulation hypothesis. Moreover, in order to maintain the sample homogeneous enough in order for it to be as representative as possible of the population of healthy fetuses, high risk pregnancies were excluded, making it impossible to confirm whether different conditions associated with stress levels might also be related with yawning frequencies. Further studies are needed to address these limitations, and confirm whether our findings are actually indicative of a stress-related modulation. Furthermore, the limited statistical power afforded by the relatively small sample of this study might have resulted in an inability to detect subtle developmental trends. Nonetheless, the results allow us to rule out a sharp variation in yawning frequencies within the considered GA window.

In summary, this study provides a significant contribution to the field of fetal behavioral research. By employing a rigorous and specific coding method for identifying yawns among general mouth openings, we have helped clarify a long-standing methodological misconception regarding both the developmental trends and the true frequency rates of fetal yawning. Our findings—showing a median frequency of around two yawns per hour that is stable across gestation and a negative association between yawning frequency and birth weight—strongly align fetal yawning with the established behavioral patterns of neonates and infants. This methodological rigor, combined with the novel finding of a seeming stress-related modulation in healthy fetuses, makes this arguably the most robust assessment of this phenomenon to date. These insights not only open new avenues for the clinical assessment of fetal neurobehavioral status but also compel continuing evaluation of the theorized functions of yawning across ontogeny as well as phylogeny, suggesting the presence in fetuses of a modulatory dynamic closer to extra-uterine life than previously understood.

## Supporting information

**S1 File. Dataset.**
(CSV)

**S2 File. R code for data analysis.**
(R)

**S3 File. Data dictionary.**
(TXT)

**S4 File. Statistical Analysis Output Log.**
(PDF)

## Acknowledgments

We extend our sincere gratitude to the dedicated staff of the Obstetrics and Gynecology Unit at the Padua Hospital for their invaluable support and assistance throughout this study. We are also deeply grateful to the mothers who generously volunteered their time and participation, without whom this research would not have been possible.

## Author contributions

**Conceptualization:** Damiano Menin, Marco Dondi.

**Data curation:** Damiano Menin.

**Formal analysis:** Damiano Menin.

**Investigation:** Paola Veronese, Maria Teresa Gervasi, Marco Dondi.

**Methodology:** Damiano Menin, Paola Veronese, Maria Teresa Gervasi, Harriet Oster, Marco Dondi.

**Supervision:** Paola Veronese, Maria Teresa Gervasi, Harriet Oster, Marco Dondi.

**Visualization:** Damiano Menin.

**Writing – original draft:** Damiano Menin, Marco Dondi.

**Writing – review & editing:** Damiano Menin, Paola Veronese, Maria Teresa Gervasi, Harriet Oster, Marco Dondi.

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
