## [Decision Letter · Decision Letter 0]

13 Oct 2025

Dear Dr. Dondi,

We look forward to receiving your revised manuscript.

Kind regards,

Andrew C Gallup, Ph.D.

Academic Editor

PLOS ONE

Journal Requirements:

2. We note that there is identifying data in the Supporting Information file  “dataset.csv”. Due to the inclusion of these potentially identifying data, we have removed this file from your file inventory. Prior to sharing human research participant data, authors should consult with an ethics committee to ensure data are shared in accordance with participant consent and all applicable local laws.

-Location data

Additional Editor Comments:

**Comments to the Author**

1. Is the manuscript technically sound, and do the data support the conclusions?

Reviewer #1: Yes

Reviewer #2: Yes

2. Has the statistical analysis been performed appropriately and rigorously?

Reviewer #1: Yes

Reviewer #2: Yes

3. Have the authors made all data underlying the findings in their manuscript fully available?

Reviewer #1: Yes

Reviewer #2: Yes

4. Is the manuscript presented in an intelligible fashion and written in standard English?

Reviewer #1: Yes

Reviewer #2: Yes

Reviewer #1: Review of PONE-D-25-41905: Fetal yawning and non-yawning mouth openings: frequency, stability and the association with birth weight.

Summary of ms. This ms presents data on human fetal yawns, using the most well-defined identification method available, FACS and Baby FACS, in order to investigate whether fetal yawn frequency as well as no-yawning mouth opening frequency are in fact related to gestational age, as found in some but not all previous studies. A second goal related to that was to explain the discrepancies of previous studies as confusion of yawn and non-yawn mouth openings. Finally, in order to investigate ways in which fetal yawning can be related to the temperature regulation/brain cooling hypotheses for post-natal homeotherms including humans, yawning frequency was related to birth weight as a possible signal of intra-uterine stress, given that "stress" is a correlate of yawning in some animal and human studies. The ms presents evidence that, indeed, carefully identified yawns were unrelated in frequency to gestational age over the weeks tested (ca. 23-32 weeks), while non-yawn mouth opening was. The data also suggest that confusing the yawn and non-yawn mouth openings, especially the long duration non-yawn openings, would have led to a relationship with gestational age. Finally, the data support a negative relationship between yawn frequency and later birth weight for the study babies, all born full-term and within the range of "normal" birth weight. These results are discussed both in terms of the pragmatic problem of identifying true yawns leading to incorrect conclusions and in terms of explaining yawning in fetal life, when it cannot contribute to actual brain cooling.

Overall, I found this study to be carefully and usefully presented, with the results described accurately and the figures very useful in interpreting the statistical statements. The discussion was handled well, given that there were really three distinct goals: to reconcile previous conflicting results on the relationship of gestational age to yawns vs non-yawn mouth openings; to investigate yawning vs birth weight as an indicator of some hard-to-detect stressors; and to place fetal yawning patterns against those hypothesized for post-natal yawning, i.e., evolutionary (or functional) explanations for yawning.

As someone familiar with non-human yawning patterns, but not an obstetrician or neonatologist, the results and discussion raised a few unanswered questions that would bear mentioning. While the in-utero environment does not allow independent cooling, differences in temperature MIGHT nevertheless elicit yawning in a developing fetus. Were the body temperatures of mothers taken during this period? And was the fetal heart rate monitored? I imagine that activity and fetal metabolism does create increases in fetal body temperature, even if damped by the amniotic fluid surround. If there were any relationships here, it might indicate early fetal thermoregulatory behavior through yawning. That it doesn't actually function might indicate simply that the feedback loop that limits yawning in adult humans or birds in hot environments is not yet operational.

Taking the opposite tack, I note from the graph that the negative relationship must be heavily influenced by both one high yawn data point in lower weight babies and strongly, the apparent zero yawns by five large (>3.5kg) infants. This seems odd to me, although there are other babies with zero or near-zero yawns. If I were to interpret the graph all by itself, I would suggest that there was something unusual about future high-birth-weight babies who infrequently yawned. (I do see from the csv file that they were in no way older at the time of the sonograms--as occurred to me.) But perhaps the sample size is an issue, giving undue weight to a few data points, and the authors clearly mentioned that as a reason for more such studies. If so or in any case, I would strongly suggest that direct temperature indicators (mom's temperature, baby heart rate, activity) be included. If they are available in this study, maybe they could be included and checked for influence relatively easily.

A few further points about the Methods, data and analysis:

I immediately wondered about fetal sex differences, especially since in my experience they influence weight, growth patterns and even average body temperatures. When I checked the R code, I see that sex was included in the models and presumably then did not influence yawning frequency in and of itself. But that should be stated in the Methods.

In addition, I would like to see the models given in the text rather than requiring the reader to call up an R-screen. Finally, please, please, please, when a complete data set is included in the supplement, help readers, reviewers and future others by including a "read-me" file that clearly defines the variables by name and their units as shown in the csv file.

I should also point out that the csv file contains a column that appears to be surnames, perhaps of the mothers. If so, that seems to leave personal identification too open. It would be better to have those coded in some way. They are not needed for the data set to be analyzed.

How did you handle "yawns as a percent of all mouth openings" in cases when no yawns were recorded for a given fetus?

Methodologically, where there were disagreements between coders, how was the final number of open-mouths and yawns decided?

Small editorial items

a. Seem to be missing a point for a 3.5 gm infant in Fig 2. I did not check any others.

b. The formatting of references vis a vis capitalization is erratic. But easily fixed of course.

c. When giving p-values, usually I see a leading zero i.e. 0.04. Maybe it is PONE practice to do otherwise. Here all are given without the zero.

d. Phylogeny is misspelled as philogeny several times in the Introduction.

e. In the methods, it would be helpful to say what is considered normal weight range for full term infants, in what ethnicity. I assume that the ethnicity of the mothers-infants, which I know does make a difference in birth weight range at least, was similar across subjects and thus did not have to be adjusted.

Reviewer #2: General comments

Overall, this is a very interesting and carefully conducted study, even if it may not be entirely groundbreaking. The topic of fetal yawning is engaging, and the authors’ attention to methodological detail is commendable. That said, the introduction feels quite long and dense in places. Personally, I find that presenting the hypotheses separately from the main flow of the introduction can interrupt the narrative, and a more integrated approach might help the reader follow the reasoning from theory to research questions more smoothly.

The discussion provides a clear summary of the findings, but it could be made more engaging. At times, the results are presented without extensive connection to existing literature, which makes it harder to fully appreciate their relevance. Additionally, ending the discussion by emphasizing study limitations tends to overshadow the contributions of the work. Highlighting the study’s strengths and the insights it provides, while still acknowledging limitations in a balanced and modest way, could help the discussion leave a stronger impression.

TITLE - The title is clear and informative, giving a good sense of the main focus of the study. It accurately reflects the variables examined, fetal yawning, non-yawning mouth openings, and birth weight, as well as the key aspects investigated, such as frequency and stability. One minor point to consider is that it reads a bit long and technical; simplifying it slightly or making it more fluid could make it even more accessible to a broader readership, without losing specificity.

Introduction

The introduction provides a very thorough overview of the main theories on yawning and does a good job of integrating both phylogenetic and ontogenetic perspectives. At the same time, it feels quite dense and occasionally repetitive, with many citations clustered together. Some sections lean more toward summarizing the literature than synthesizing it into a coherent argument leading to the rationale of the study. It might be helpful to clarify more explicitly why fetal yawning deserves specific attention beyond the “blind spot” of the brain cooling hypothesis.

Some statements could benefit from slightly more cautious wording. For example, the text sometimes presents the brain cooling hypothesis as a well-established or “unifying” explanation, when it is still a topic of active debate. A softer tone here would better reflect the ongoing discussion in the field. Additionally, the transition from broad theories about yawning in adults to observations in fetuses could be made smoother, to help the reader understand the link between macro-level and micro-level perspectives.

In the section on modulatory factors (lines 45–55), the numerous examples listed (stress, hunger, pain, arousal, thermoregulation) could be more clearly distinguished between well-supported empirical findings and more speculative associations, to avoid overinflating claims.

The discussion of the brain cooling hypothesis (lines 60–79) is very thorough, but could benefit from a more balanced approach. At present, it emphasizes supporting evidence while downplaying limitations, such as the difficulty of measuring brain temperature changes in non-rodent species. Phrases like “unparalleled explanatory and predictive power” might be softened to present the strengths of the hypothesis without overstating them.

The section on contagious yawning (lines 89–101) is relevant, but it could more clearly separate proximate and ultimate explanations. While the Attentional Bias and Emotional Bias hypotheses are mentioned, their connection to evolutionary theory or the broader framework of yawning could be more explicitly clarified. A reference to recent work on zebrafish could also strengthen this section.

The discussion of yawning in poikilotherms (lines 102–111) adds an interesting phylogenetic perspective. Some of the interpretations, such as suggesting that yawning in fish reflects “primitive functions,” could be presented more tentatively, acknowledging that this remains a hypothesis rather than established fact.

The section on fetal yawning (lines 112–124) introduces the core topic well, but the logical flow could be clearer. Presenting Walusinski’s critique followed immediately by Gallup & Eldakar’s counterpoint leaves the reader uncertain about what is empirically known. Phrasing like “does not falsify the brain cooling hypothesis” could be reframed more positively as an opportunity to refine and extend the theory.

The discussion of empirical variability (lines 125–155) is one of the stronger parts of the introduction, as it addresses methodological inconsistencies in prior studies. However, this comes relatively late in the text, after a substantial theoretical discussion. Bringing these points earlier could help the reader understand why the present study is needed. Additionally, the critique of Reissland et al. could be softened by acknowledging their pioneering role, even while highlighting the methodological improvements of later studies.

Methods

The sample size and gestational age range (n = 32 fetuses, 23–31 weeks) are reasonable given practical constraints, but it is worth noting that the study may have limited power to detect subtle developmental trends, which should be considered when interpreting hypotheses H2–H4.

Discussion

The discussion clearly follows the results but could benefit from a more engaging narrative. At present, it mostly restates findings without fully connecting them to broader theoretical perspectives, such as the brain cooling hypothesis or the evolutionary and developmental context of yawning. Interpretations, for example regarding the negative association between yawning frequency and birth weight, are thoughtfully proposed but remain somewhat ambiguous. It might help to guide the reader toward which explanations are more plausible, while maintaining appropriate caution.

The focus on methodological validity is commendable, but the discussion could further highlight the study’s contribution. Ending with limitations, while important, may leave the reader without a clear sense of the significance of the findings. Expanding briefly on implications for fetal neurobehavioral assessment or the study of yawning modulation across species could strengthen the impact.

Overall, the study appears methodologically sound and makes an interesting contribution, but the discussion would be enhanced by a tighter integration of results with theory, clearer prioritization of interpretations, and a more conclusive closing statement that emphasizes what the study adds to the field.

**Do you want your identity to be public for this peer review?** For information about this choice, including consent withdrawal, please see our Privacy Policy

Reviewer #1: **Yes:** Anne B. Clark, PhD

Reviewer #2: No

---

## [Author Response · Author response to Decision Letter 1]

27 Nov 2025

Reviewer #1: Review of PONE-D-25-41905: Fetal yawning and non-yawning mouth openings: frequency, stability and the association with birth weight.

Summary of ms. This ms presents data on human fetal yawns, using the most well-defined identification method available, FACS and Baby FACS, in order to investigate whether fetal yawn frequency as well as no-yawning mouth opening frequency are in fact related to gestational age, as found in some but not all previous studies. A second goal related to that was to explain the discrepancies of previous studies as confusion of yawn and non-yawn mouth openings. Finally, in order to investigate ways in which fetal yawning can be related to the temperature regulation/brain cooling hypotheses for post-natal homeotherms including humans, yawning frequency was related to birth weight as a possible signal of intra-uterine stress, given that "stress" is a correlate of yawning in some animal and human studies. The ms presents evidence that, indeed, carefully identified yawns were unrelated in frequency to gestational age over the weeks tested (ca. 23-32 weeks), while non-yawn mouth opening was. The data also suggest that confusing the yawn and non-yawn mouth openings, especially the long duration non-yawn openings, would have led to a relationship with gestational age. Finally, the data support a negative relationship between yawn frequency and later birth weight for the study babies, all born full-term and within the range of "normal" birth weight. These results are discussed both in terms of the pragmatic problem of identifying true yawns leading to incorrect conclusions and in terms of explaining yawning in fetal life, when it cannot contribute to actual brain cooling.

Overall, I found this study to be carefully and usefully presented, with the results described accurately and the figures very useful in interpreting the statistical statements. The discussion was handled well, given that there were really three distinct goals: to reconcile previous conflicting results on the relationship of gestational age to yawns vs non-yawn mouth openings; to investigate yawning vs birth weight as an indicator of some hard-to-detect stressors; and to place fetal yawning patterns against those hypothesized for post-natal yawning, i.e., evolutionary (or functional) explanations for yawning.

Q1: As someone familiar with non-human yawning patterns, but not an obstetrician or neonatologist, the results and discussion raised a few unanswered questions that would bear mentioning. While the in-utero environment does not allow independent cooling, differences in temperature MIGHT nevertheless elicit yawning in a developing fetus. Were the body temperatures of mothers taken during this period? And was the fetal heart rate monitored? I imagine that activity and fetal metabolism does create increases in fetal body temperature, even if damped by the amniotic fluid surround. If there were any relationships here, it might indicate early fetal thermoregulatory behavior through yawning. That it doesn't actually function might indicate simply that the feedback loop that limits yawning in adult humans or birds in hot environments is not yet operational.

R1: This is a very interesting idea, that we hope to be able to investigate in future studies (we also think a similar approach would be interesting in fish or other poichiloterms, although that falls outside of our expertise). Unfortunately, however, neither body temperatures or fetal heart rate were monitored. We have added this as a limitation in the final paragraph of the discussion:

“It should be noted as a limitation that this study did not employ experimental manipulation or selective observation of fetuses under specific conditions, and did not include measures (e.g., maternal body temperatures or fetal heart rate) that could be useful in disentangling the effects of different modulatory mechanisms, with particular regard for the thermoregulation hypothesis. Moreover, in order to maintain the sample homogeneous enough in order for it to be as representative as possible of the population of healthy fetuses, high risk pregnancies were excluded, making it impossible to confirm whether different conditions associated with stress levels might also be related with yawning frequencies. Further studies are needed to address these limitations, and confirm whether our findings are actually indicative of a stress-related modulation.”

Q2: Taking the opposite tack, I note from the graph that the negative relationship must be heavily influenced by both one high yawn data point in lower weight babies and strongly, the apparent zero yawns by five large (>3.5kg) infants. This seems odd to me, although there are other babies with zero or near-zero yawns. If I were to interpret the graph all by itself, I would suggest that there was something unusual about future high-birth-weight babies who infrequently yawned. (I do see from the csv file that they were in no way older at the time of the sonograms--as occurred to me.) But perhaps the sample size is an issue, giving undue weight to a few data points, and the authors clearly mentioned that as a reason for more such studies. If so or in any case, I would strongly suggest that direct temperature indicators (mom's temperature, baby heart rate, activity) be included. If they are available in this study, maybe they could be included and checked for influence relatively easily.

R2: We agree with the reviewer’s assessment but, unfortunately, for this study these data weren’t available, as they are not routinely acquired during fetal ultrasound scans. Regarding the fetuses with no yawns, please consider that, because yawning is a relatively low-frequency behavior and the observation time is limited because of the nature of fetal ultrasound scans, frequencies being zero doesn’t mean the “true value” for the yawning rate of these fetuses is 0 yawns per minute. However, Poisson models are designed to handle a moderate number of zeros in the dependent variable, and zero-inflated models were not a fit for this dataset, so we maintain this was the correct approach to analyze this dataset, although other studies are needed to confirm in particular the association of yawning rates and birthweight.

A few further points about the Methods, data and analysis:

Q3: I immediately wondered about fetal sex differences, especially since in my experience they influence weight, growth patterns and even average body temperatures. When I checked the R code, I see that sex was included in the models and presumably then did not influence yawning frequency in and of itself. But that should be stated in the Methods.

R3: As per the reviewer’s intuition, sex was initially introduced as a predictor (as we are well aware of the evidences and discussion about gender differences in yawning), but we decided to exclude it from the final version of the models. As the reviewer anticipated, in fact, sex did not show any association with the study variables, nor its introduction affected in any significant way the models (both in terms of fit and of other effects). Moreover, although sex/gender, can influence yawning rates under particular conditions even few months after birth, we are not aware of any study highlighting sex differences in fetal behavior (when sexual differentiation, prior to mini-puberty observed during the first postnatal months, is minimal). In order to make this clear, we introduced a specification in the methods:

“Although fetal sex was included in preliminary analyses as a predictor, it was dropped from the final regressions because it did not significantly improve the model fit (tested via ANOVA) and showed no significant correlation with other fixed effects. This decision is further supported by the current lack of evidence for sex-based modulation of behavior in fetuses, a population showing minimal sexual differentiation. The complete results of the full models and the ANOVA, however, are provided in the Supporting Information.”

Q4: In addition, I would like to see the models given in the text rather than requiring the reader to call up an R-screen. Finally, please, please, please, when a complete data set is included in the supplement, help readers, reviewers and future others by including a "read-me" file that clearly defines the variables by name and their units as shown in the csv file.

R4: The R output was pasted in a new file (log.pdf), so that readers can access the results for both main models and those with sex as a covariate (and the ANOVA). A read-me file was also created and uploaded (readme.txt), explaining the variables included in the dataset.

Q5: I should also point out that the csv file contains a column that appears to be surnames, perhaps of the mothers. If so, that seems to leave personal identification too open. It would be better to have those coded in some way. They are not needed for the data set to be analyzed.

R5: We sincerely want to thank the reviewer for taking the time to carefully examine even the supplementary materials, picking up on this important point. The database is now fully anonimized.

Q6: How did you handle "yawns as a percent of all mouth openings" in cases when no yawns were recorded for a given fetus?

R6: Because this metric represents the percentage of mouth openings (including yawns and non-yawn MOs) that were yawns (calculated as yawns*100/yawns+non-yawn mouth openings), this case is pretty straightforward, and results in the value being set to zero. In the few cases where both yawns an non-yawn mouth openings were zero, this variable was set to NA.

Q7: Methodologically, where there were disagreements between coders, how was the final number of open-mouths and yawns decided?

R7: We used a method that is commonly adopted in the study of perinatal and infant behavior, as detailed in the Data analysis section. In particular, for both behavioral patterns considered, we adopted a one-second threshold for defining agreements and disagreements. This means that, when coder 1 identified e.g. one yawn with the offset less than one second away from a yawn identified by coder 2, that was considered a positive agreement (1:1). If one coder did not identify one event that the other did, that was considered as a disagreement (1:0 or 0:1). Finally, the remaining observation time, divided by twice the threshold (2 s) was considered as the number of negative agreements. Having built such a contingency table, we calculated Cohen’s Kappa, a statistical index for assessing the agreement between two raters in a classification task, which takes into account the agreement expected at random. This analysis shows that the agreement in the individuation of yawns was perfect (Kappa = 1), while there were some disagreements in the identification of mouth openings, that showed however more than satisfactory reliability (Kappa = .88, the threshold generally used is .7 or even .6). Overall, the reliability proved to be very good, therefore the data acquired by coder 1 was used for statistical analyses, without need for retraining or other interventions.

Q8: Seem to be missing a point for a 3.5 gm infant in Fig 2. I did not check any others.

R8: The two points are actually coincident (same number of events over the same observation time). This is shown by ggplot2 using a slightly different color (brown-ish)

Q9: The formatting of references vis a vis capitalization is erratic. But easily fixed of course.

R9: This inconsinstency in capitalization was fixed (only the first word or the title was capitalized, in lines with PONE guidelines.

Q10: When giving p-values, usually I see a leading zero i.e. 0.04. Maybe it is PONE practice to do otherwise. Here all are given without the zero.

R10: Because we found no indication about this in the author guidelines, we adopted the APA rule about leading zeros, i.e, to only use them when the coefficient reported can exceed the value of 1. With p and Cohen’s Kappa, therefore, no leading zero was used. Of course we are open to changes if this goes against PONE practices.

Q11: Phylogeny is misspelled as philogeny several times in the Introduction.

R11: Thanks for the catch, that was corrected.

Q12: In the methods, it would be helpful to say what is considered normal weight range for full term infants, in what ethnicity. I assume that the ethnicity of the mothers-infants, which I know does make a difference in birth weight range at least, was similar across subjects and thus did not have to be adjusted.

R12: The reviewer is right in implying that different ethnicities might have different tipical ranges for birth weight, although the 2.5-4.0 kg range is generally applied to neonates of any backgound. We introduced this information in the Participants section:

“All mothers were of Italian nationality, with a self-reported Italian ethnic background.”

Reviewer #2: General comments

Q1: Overall, this is a very interesting and carefully conducted study, even if it may not be entirely groundbreaking. The topic of fetal yawning is engaging, and the authors’ attention to methodological detail is commendable. That said, the introduction feels quite long and dense in places. Personally, I find that presenting the hypotheses separately from the main flow of the introduction can interrupt the narrative, and a more integrated approach might help the reader follow the reasoning from theory to research questions more smoothly.

R1: We sincerely thank the reviewer for this constructive stylistic suggestion. We agree that integrating the hypotheses more smoothly within the main text is generally preferred for narrative flow. However, due to the specific complexity and multi-faceted nature of our five hypotheses, which address distinct methodological and developmental issues, we believe that presenting them schematically and distinctly offers maximum clarity to the reader. This structure allows readers to quickly locate and reference the precise research question and prediction being discussed in the results and discussion sections, which we feel outweighs the minor interruption to the narrative flow

Q2: The discussion provides a clear summary of the findings, but it could be made more engaging. At times, the results are presented without extensive connection to existing literature, which makes it harder to fully appreciate their relevance. Additionally, ending the discussion by emphasizing study limitations tends to overshadow the contributions of the work. Highlighting the study’s strengths and the insights it provides, while still acknowledging limitations in a balanced and modest way, could help the discussion leave a stronger impression.

R3: We tried to make the discussion more engaging and to emphasize the study’s strengths and contribution by adding a final paragraph to the discussion (after the limitations), providing a general synopsis of the most important results and of their relevance (see below).

“In summary, this study provides a significant contribution to the field of fetal behavioral research. By employing a rigorous and specific coding method for identifying yawns among general mouth openings, we have helped clarify a long-standing methodological misconception regarding both the developmental trends and the true frequency rates of fetal yawning. Our findings—showing a median frequency of around two yawns per hour that is stable across gestation and a negative association between yawning frequency and birth weight—strongly align fetal yawning with the established behavioral patterns of neonates and infants. This methodological rigor, combined with the novel finding of a seeming stress-related modulation in healthy fetuses, makes this arguably the most methodologically rigorous study of this phenomenon to date. These insights not only open new avenues for the clinical assessment of fetal neurobehavioral status but also compel a re-evaluation of the theoretical functions of yawning, suggesting a modulatory dynamic closer to extra-uterine life than previously understood.”

Q4: TITLE - The title is clear and informative, giving a good sense of the main focus of the study. It accurately reflects the variables examined, fetal yawning, non-yawning mouth openings, and birth weig

---

## [Decision Letter · Decision Letter 1]

23 Dec 2025

Dear Dr. Dondi,

Thank you for submitting your manuscript to PLOS ONE. After careful consideration, I feel that it has merit but does not fully meet PLOS ONE’s publication criteria as it currently stands. Therefore, I invite you to submit a revised version of the manuscript that addresses the points raised below.

We look forward to receiving your revised manuscript.

Kind regards,

Andrew C Gallup, Ph.D.

Academic Editor

PLOS One

Journal Requirements:

Reviewers' comments:

Reviewer's Responses to Questions

**Comments to the Author**

Reviewer #1: (No Response)

Reviewer #2: All comments have been addressed

2. Is the manuscript technically sound, and do the data support the conclusions?

Reviewer #1: Yes

Reviewer #2: Yes

3. Has the statistical analysis been performed appropriately and rigorously?

Reviewer #1: Yes

Reviewer #2: Yes

4. Have the authors made all data underlying the findings in their manuscript fully available?

Reviewer #1: Yes

Reviewer #2: Yes

5. Is the manuscript presented in an intelligible fashion and written in standard English?

Reviewer #1: Yes

Reviewer #2: Yes

Reviewer #1: In reviewing the authors' replies to both reviewers' points, I think the additions and alterations that the authors have made are adequate and reasonable. If anything, the final paragraph (changed in answer to Rev 2) might be a little too effusive or "expansive" on the yawning conclusions, e.g. "compel a reevaluation of the theoretical functions of yawning". I might have phrased this as "compel continuing evaluation of the theorized functions of yawning across ontogeny as well as phylogeny", which has clearly been relatively neglected.

However, one cannot never write for the many different readers from different backgrounds in making fine points.

To the mind of this reviewer, the authors have made good use of both reviews and have improved the paper where criticism was given. I will be very interested in follow-up studies where "stress" has a stronger proxy, as well as where the temperature of the uterine environment and the current metabolic output of the fetus might be measured.

With respect to editing, there are a few places where grammar fixes could be made. For instance,

a) Ref 1 has two "." at end of title and the species names are not italicized or underlined. (Maybe they are not in original title...I forget.)

b) Repetitive use of "methodological rigor" in ll 380-381

c) l. 371 "nonetheless" is incorrectly used as a conjunction. But mostly it is stylistically fine.

Reviewer #2: Thank you for addressing all my comments thoroughly and clearly. The revisions you implemented fully resolve the issues I had raised and substantially improve the manuscript. I have no further remarks and am satisfied with the final version, supporting the acceptance of the paper in its current form.

**Do you want your identity to be public for this peer review?** For information about this choice, including consent withdrawal, please see our Privacy Policy

Reviewer #1: **Yes:** Anne B. Clark, Ph.D.

Reviewer #2: No

---

## [Author Response · Author response to Decision Letter 2]

5 Jan 2026

Response to Academic Editor and Reviewer

Academic Editor’s comments

Q1: In my evaluation of the revised manuscript, I noticed an error in reference to the existing literature that should be addressed. On page 4, lines 73-75, it is stated that brain temperature modulation following yawning has only been documented in rats when it has been reported in at least two other species as well. The first is budgerigars (Melopsittacus undulatus), and the second, most recently, is mice. I have provided these references below, which should be included for accuracy in the final version.

Gallup, A. C., Herron, E., Militello, J., Swartwood, L., Cortes, C., & Eguibar, J. R. (2017). Thermal imaging reveals sizable shifts in facial temperature surrounding yawning in budgerigars (Melopsittacus undulatus). Temperature, 4(4), 429-435.

Alam, M. T., Ahasan, M. M., Shimizu, S., Murata, Y., Taniguchi, M., & Yamaguchi, M. (2025). Differential potentiation of odor aversion and yawning by melanocortin 4 receptor signaling in distinct regions of the ventral striatum. Frontiers in Neuroscience, 19, 1668410.

R1: We thank the Academic Editor for spotting this oversight. We have included these citations and the corresponding information in the revised introduction (see below):

“It is important to note, however, that evidence of this modulation is currently available only for rats (25), budgerigars (26) and mice (27), and additional studies are needed to confirm these results in other species.”

Reviewers' comments

Reviewer #1:

Q1: In reviewing the authors' replies to both reviewers' points, I think the additions and alterations that the authors have made are adequate and reasonable. If anything, the final paragraph (changed in answer to Rev 2) might be a little too effusive or "expansive" on the yawning conclusions, e.g. "compel a reevaluation of the theoretical functions of yawning". I might have phrased this as "compel continuing evaluation of the theorized functions of yawning across ontogeny as well as phylogeny", which has clearly been relatively neglected.

However, one cannot never write for the many different readers from different backgrounds in making fine points.

R1: We agree with the reviewer’s assessment and very much appreciate her suggestion for an alternative, more balanced phrasing, which we have adopted in the revised version (see below).

“These insights not only open new avenues for the clinical assessment of fetal neurobehavioral status but also compel continuing evaluation of the theorized functions of yawning across ontogeny as well as phylogeny, suggesting the presence in fetuses of a modulatory dynamic closer to extra-uterine life than previously understood.”

Q2: To the mind of this reviewer, the authors have made good use of both reviews and have improved the paper where criticism was given. I will be very interested in follow-up studies where "stress" has a stronger proxy, as well as where the temperature of the uterine environment and the current metabolic output of the fetus might be measured.

R2: We wish to deeply thank the reviewer for the genuinely helpful criticism and support she provided. We hope to be able in the future to carry out studies suited to test more directly the hypothesis of a stress-related yawning modulation in fetuses, or that this study can inspire other scholars to conduct similar studies.

Q3: With respect to editing, there are a few places where grammar fixes could be made. For instance,

a) Ref 1 has two "." at end of title and the species names are not italicized or underlined. (Maybe they are not in original title...I forget.)

b) Repetitive use of "methodological rigor" in ll 380-381

c) l. 371 "nonetheless" is incorrectly used as a conjunction. But mostly it is stylistically fine.

R3: Thanks for these comments, we made the fixes in the revised version. In particular:

a) The title was corrected, removing the dot and italicizing the species names;

b) “Methodologically rigorous study” was replaced with “robust assessment”, in order to avoid the repetition. The sentence now reads “ This methodological rigor, combined with the novel finding of a seeming stress-related modulation in healthy fetuses, makes this arguably the most robust assessment of this phenomenon to date”

c) We have corrected the sentence by replacing the comma with a period and starting a new sentence with "Nonetheless" (now used correctly as a sentence connector rather than a conjunction).

Reviewer #2:

Q1: Thank you for addressing all my comments thoroughly and clearly. The revisions you implemented fully resolve the issues I had raised and substantially improve the manuscript. I have no further remarks and am satisfied with the final version, supporting the acceptance of the paper in its current form.

R1: We wish to thank the reviewer for their constructive criticism and for their positive final evaluation of our work. We are pleased that the revisions have addressed all concerns and that the manuscript is now considered suitable for publication.

---

## [Editor Report · Decision Letter 2]

6 Jan 2026

Fetal yawning and mouth openings: frequency, developmental trends, and association with birth weight

PONE-D-25-41905R2

Dear Dr. Dondi,

We’re pleased to inform you that your manuscript has been judged scientifically suitable for publication and will be formally accepted for publication once it meets all outstanding technical requirements.

Kind regards,

Andrew C Gallup, Ph.D.

Academic Editor

PLOS One
---

## [Editor Report · Acceptance letter]

PONE-D-25-41905R2

PLOS One

Dear Dr. Dondi,

I'm pleased to inform you that your manuscript has been deemed suitable for publication in PLOS One. Congratulations! Your manuscript is now being handed over to our production team.

Kind regards,

on behalf of

Andrew C Gallup

Academic Editor

PLOS One